



# Impact of agricultural emission reductions on fine particulate matter and public health.

Andrea Pozzer[1], Alexandra P. Tsimpidi[1], Vlassis A. Karydis[1], Alexander de Meij[2,*], and Jos Lelieveld[1,3]

[1]Max Planck Institute for Chemistry, Mainz, Germany
[2]Noveltis, Sustainable Development, Rue du Lac, F-31670 Labege, France
[3]The Cyprus Institute, Nicosia, Cyprus
[*]now at: MetClim, Varese, Italy

*Correspondence to:* A.Pozzer,andrea.pozzer@mpic.de

**Abstract.** A global chemistry-climate model has been used to study the impacts of pollutants released by agriculture on fine particulate matter ($PM_{2.5}$), with a focus on Europe, North America and East Asia. Simulations reveal that a relatively strong reduction in $PM_{2.5}$ levels can be achieved by decreasing agricultural emissions, notably of ammonia ($NH_3$), released from fertilizer use and animal husbandry. The absolute impact on $PM_{2.5}$ reduction is strongest in East Asia, even for small emission

decreases. Conversely, over Europe and North America, aerosol formation is not directly limited by the availability of ammonia. Nevertheless, reduction of $NH_3$ can also substantially decrease $PM_{2.5}$ concentrations over the latter regions, especially when emissions are abated systematically. Our results document how reduction of agricultural emissions decreases aerosol pH due to the depletion of aerosol ammonium, which affects particle liquid phase and heterogeneous chemistry. Further, it is shown that a 50% reduction of agricultural emissions could prevent the mortality attributable to air pollution by $\sim 250$ thousands people

per year worldwide, amounting to reductions of 30%, 19% and 8% over North America, Europe and East Asia, respectively. A theoretical 100% reduction could even reduce the number of deaths globally by about 800 thousand per year.

## 1 Introduction

Atmospheric aerosol particles are a major constituent of ambient air and have a large impact on atmospheric chemistry, clouds, radiative transfer and climate, and also induce adverse human health effects that contribute to mortality (Stocker et al., 2013;

Lelieveld et al., 2015). Particulate matter (PM) with an aerodynamic diameter smaller than 2.5 μm ($PM_{2.5}$) contributes to air pollution through intricate interactions between emissions of primary particles and gaseous precursors, photochemical transformation pathways, and meteorological processes that control transport and deposition.

As shown by Lelieveld et al. (2015) and Bauer et al. (2016), agricultural emissions play a leading role in the formation of $PM_{2.5}$ in various regions of the world, for example in Central and Eastern Europe. Agricultural emissions are mostly related

to animal husbandry and fertilizer use, and to a lesser extent also to the burning of crop residues (Aneja et al., 2008); around 10% of worldwide biomass burning emissions can be ascribed to agricultural activities (Doering et al., 2009b). The general importance of agricultural emissions for air quality was also previously identified by Zhang et al. (2008), Tsimpidi et al. (2007) and Megaritis et al. (2013).



The dominant trace gas emitted by agricultural activities is ammonia ($NH_3$). Around 80-90% of the atmospheric $NH_3$ emissions in industrialized regions are from the agricultural sector (Sotiropoulou et al., 2004; Lamarque et al., 2011; van Vuuren et al., 2011b, a). Ammonia is formed and released during the decomposition of manure and organic matter, mostly from animal farming and the associated manure processing, with an additional contribution from (synthetic) fertilizer use.

Ammonia is a toxic gas at very high concentrations, with a pungent smell that irritates the eyes and respiratory system. $NH_3$ is also a major alkaline gas in the atmosphere and plays an important role in neutralizing acids in the aerosol and cloud liquid phase, forming ammonium sulfate and ammonium nitrate (ammonium salts) (Behera et al., 2013). Therefore $NH_3$ contributes to secondary aerosol formation and the overall particulate matter burden, and decreases the acidity of the aerosols, which in turn increases the solubility of weak acids (e.g., $HCOOH$, $SO_2$). The aerosol pH plays an important role in the reactive uptake

and release of gases, which can affect ozone chemistry, particle properties such as hygroscopic growth and scattering efficiency of sunlight, and deposition processes (Zhang et al., 2007; Thornton et al., 2010; Pathak et al., 2011).

  Tsimpidi et al. (2007) showed that a 50% reduction of $NH_3$ emissions leads to a 4% and 9% decrease in $PM_{2.5}$ over the Eastern USA in July and January, respectively. The reduction of $NH_3$ emissions was found to be the most effective $PM_{2.5}$ control measure for the winter period over the Eastern USA, compared to similar reductions of $SO_2$, $NO_x$ and VOC emissions

(Pinder et al., 2008; Tsimpidi et al., 2007, 2008; Karydis et al., 2011). Megaritis et al. (2013) and Bessagnet et al. (2014) found that over Europe the reduction of $NH_3$ emissions is the most effective control strategy to mitigate $PM_{2.5}$, during both summer and winter, mainly due to a significant decrease of ammonium nitrate. Further, De Meij et al. (2009), showed that reducing the $NH_3$ emissions by agriculture by 50% resulted in a decrease of $PM_{2.5}$ concentrations up to 2.4 $\mu g/m^3$ over the Po Valley region (Italy). This confirms the finding of de Meij et al. (2006), who showed that for short-lived species like $NO_x$ and $NH_3$,

short-term fluctuations of the emissions play an important role in the formation of nitrate aerosol. According to Wang et al. (2011), $NH_3$ emissions contribute 8-11% to $PM_{2.5}$ concentrations in East China, which is comparable with the contributions of $SO_2$ (9-11%) and $NO_x$ (5-11%) emissions. However, the air quality benefits of controlling $NH_3$ emissions could be offset by the potential enhancement of aerosol acidity. Weber et al. (2016) showed that despite the large investments in sulfur dioxide emission reductions, the acid/base gas particle system in the southeastern USA is buffered by the partitioning of semivolatile

$NH_3$, making the pH insensitive to $SO_2$ controls. Several studies have been performed on the impact of $NH_3$ on aerosol nitrate (Pye et al., 2009; Heald et al., 2012; Schaap et al., 2004; Pinder et al., 2007; Holt et al., 2015), and sulfate (Redington et al., 2009; Paulot et al., 2016; Wang et al., 2011), mostly with a regional rather than a global view.

  As $PM_{2.5}$ has been clearly associated with many health impacts, including acute lower respiratory infections (ALRI), cerebrovascular disease (CEV), ischaemic heart disease (IHD), chronic obstructive pulmonary disease (COPD) and lung cancer

(LC) (Burnett et al., 2014). Due to its strong contribution to the $PM_{2.5}$ mass, control strategies in $NH_3$ emissions could possibly reduce the mortality attributable to air pollution. Such analysis has been performed regionally for Europe (Brandt et al., 2013) and the U.S.A. (Paulot and Jacob, 2014; Muller and Mendelsohn, 2007). Further, a detailed analysis on the global scale was performed by Lee et al. (2015) who showed the importance of ammonia as a contributor to mortality attributable to air pollution. Nevertheless, Lee et al. (2015) assumed an ammonia reduction of 10%, and the health effects were linearized around

the present-day concentrations. As the exposure-response functions, linking $PM_{2.5}$ to mortality attributable to air pollution,





are highly non-linear at relatively low concentrations, the mortality reduction estimation could change drastically for strong reductions of ammonia emissions. Therefore, in this work, more aggressive reductions are studied (see Sect.2).

Furthermore, there is a need to investigate the impact of $NH_3$ emission reductions not only on $PM_{2.5}$ concentrations, but also account for particle acidity and aerosol composition. The goal of this work is to understand the impact of global agricultural

emissions on model simulated $PM_{2.5}$ concentrations, the effects on aerosol pH and the potential consequences for human health, with a focus on four continental regions where air quality limits and guidelines for $PM_{2.5}$ are often exceeded, i.e., North America, Europe, South and East Asia. North America is defined as the region that encompasses the U.S.A and Canada, Europe is represented by the European continent (including Turkey) excluding Russia, South Asia includes India, Sri Lanka, Pakistan, Bangladesh, Nepal and Buthan, while the East Asia region includes China, North and South Korea, and Japan (see

Fig.1).

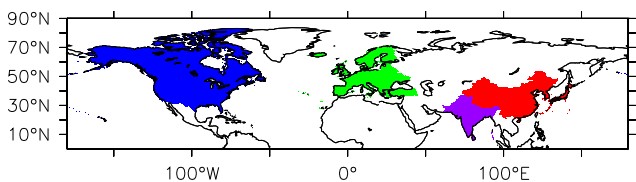

**Figure 1.** Regions addressed in this study, i.e., North America (blue), Europe (green), South Asia (yellow) and East Asia (red).

This work may also support policy actions aimed at controlling ammonia emissions, e.g., formulated in the European Union Clean Air Program (http://ec.europa.eu/environment/air/pollutants/ceilings.htm) which sets ceilings for national emissions for sulfur dioxide, nitrogen oxides, volatile organic compounds, fine particulate matter, and also for ammonia.

## 2 Methodology

In this study the EMAC (ECHAM5/MESSy Atmospheric Chemistry) model version 1.9 was used. EMAC is a combination of the general circulation model ECHAM5 (Roeckner et al., 2006, version 5.3.01) and the Modular Earth Submodel System (Jöckel et al., 2005, MESSy, version 1.9), Extensive evaluation of the model can be found in Jöckel et al. (2006); Pozzer et al. (2007, 2012a); Pringle et al. (2010a); de Meij et al. (2012a). ECHAM5 has been used at the T106L31 resolution, corresponding to a horizontal resolution of $\sim 1.1 \times 1.1°$ of the quadratic Gaussian grid, and with 31 vertical levels up to 10 hPa in the lower

stratosphere. The model set-up is the same as that presented by (Pozzer et al., 2012a, b) and is briefly summarized here. The anthropogenic emissions are for the year 2010 from the EDGAR-CIRCE (Doering et al., 2009a, c, Emission Database for Global Atmospheric Research) database, distributed vertically to account for different injection altitudes (Pozzer et al., 2009). Bulk natural aerosol emissions (i.e., desert dust and sea spray), are treated using offline monthly emissions files based on AEROCOM (Dentener et al., 2006) and hence are independent of the meteorological conditions. The atmospheric chemistry

is simulated with the MECCA (Module Efficiently Calculating the Chemistry of the Atmosphere) submodel by Sander et al. (2005, 2011), and the aerosol microphysics and gas-aerosol partitioning are calculated by the Global Modal-aerosol eXtension



**Table 1.** Summary of the comparison of model data to pseudo-observations of $PM_{2.5}$ concentrations (van Donkelaar et al., 2010). OAM and MAM are the spatial arithmetic mean of the observations and of the model results ($REF$ simulation), respectively (in µg/m$^3$), based on the annual averages. The model results were masked in locations where no observations are available. PF2 is the percentage of model results within a factor of two of the observations.

| region | MAM | OAM | MAM/OAM | PF2 |
|---|---|---|---|---|
| Europe | 9.00 | 11.96 | 0.75 | 0.95 |
| North America | 4.31 | 5.89 | 0.72 | 0.80 |
| South Asia | 24.49 | 24.95 | 0.98 | 0.95 |
| East Asia | 33.60 | 27.56 | 1.22 | 0.81 |
| World | 22.58 | 13.02 | 1.73 | 0.75 |

(GMXe) aerosol module (Pringle et al., 2010a, b). Gas / aersosol partitioning is calculated using the ISORROPIA-II model (Fountoukis and Nenes, 2007; Nenes et al., 1998a, b). Following the approach of Pozzer et al. (2012b), the year 2010 is used as reference year, and the feedback between chemistry and dynamics was switched-off, and therefore all simulations described here are based on the same (binary identical) dynamics and consequent transport of tracers.

5     The model has been evaluated against satellite based $PM_{2.5}$ estimates (van Donkelaar et al., 2010); the results are shown in Fig.2 and are summarized in Tab.1, also focusing on the four regions focus of this study (i.e., North America, Europe, South and East Asia). Compared to global satellite derived $PM_{2.5}$ concentrations this model version, with prescribed dust emissions, consistently overestimates $PM_{2.5}$ over desert areas (see Fig.2). However, the average concentration of $PM_{2.5}$ at the surface in the regions of interest is within 30% of the observations. For Europe and South Asia, 95% of the simulated $PM_{2.5}$

10   concentrations are within a factor of 2 of the observations, while for North America and East Asia this is about 80%.

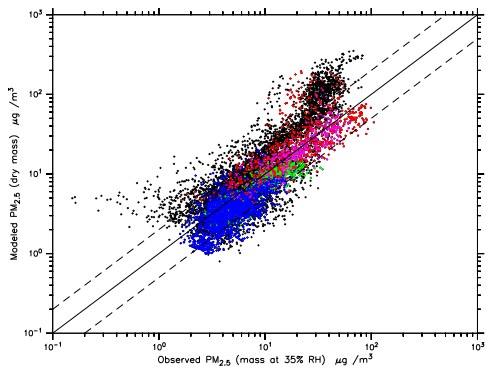

**Figure 2.** Scatter plot of observed and modeled yearly averaged concentrations of $PM_{2.5}$ (in µg/m$^3$). The colors denote the regions, i.e., blue North America, green Europe, purple East Asia and red East Asia.

Further, sulfate-ammonium-nitrate has been compared with station observations from different databases, such as from EPA (United States Environmental Protection Agency), EMEP (European Monitoring and Evaluation Programme) and EANET



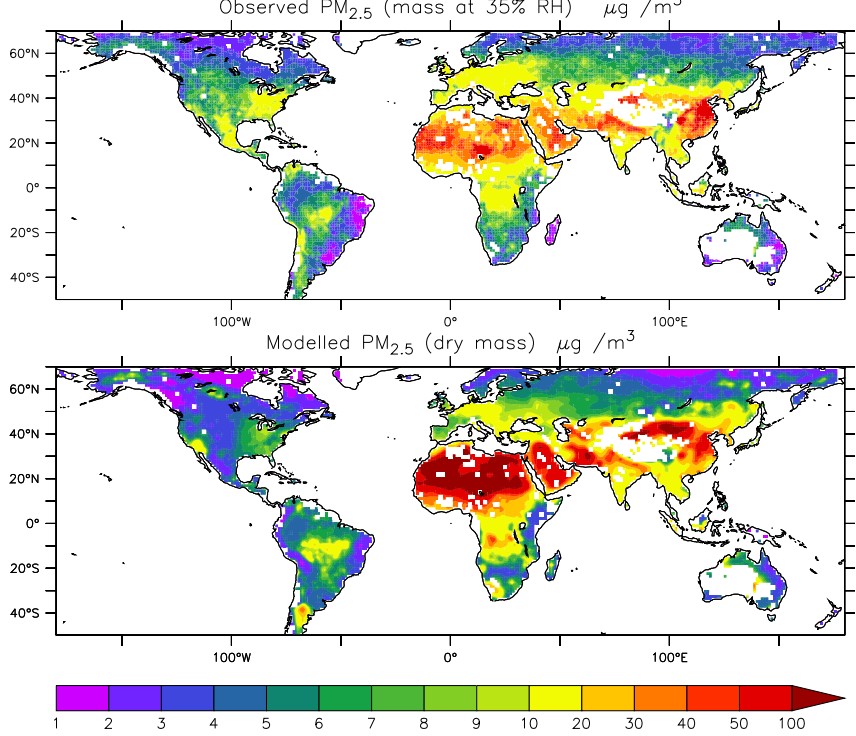

**Figure 3.** UPPER: observed annual mean $PM_{2.5}$ from (van Donkelaar et al., 2010), LOWER: simulated annual mean $PM_{2.5}$ ($REF$ simulation), both in $\mu g/m^3$.

(Acid Deposition Monitoring Network in East Asia) for the year 2010. The results are shown in Fig. 4 and summarized in Tab.2.

While sulfate is well reproduced, with more than $\sim 85\%$ of the model results within a factor of two compared to the observations, nitrate is overestimated in North America and Europe by $\sim 50\%$, although nitric acid is reproduced accurately by the model (based on comparison with observations from Emmons et al. (2000), see Jöckel et al. (2006)). As the nitrate concentrations seems to be on the high end of the observations, it must be acknowledged that the effect of reducing ammonia emissions from agriculture could be overestimated. On the other hand, nitrate predictions are in a good agreement with the measurements over East Asia. Further, ammonium concentrations are well captured by the model, with more than 75% of the modeled results being within a factor of 2 compared to observations. Further evaluation can be found in Pozzer et al. (2012a, b) and de Meij et al. (2012b).

In the current analysis four simulations with the EMAC model are used to study the impacts on $PM_{2.5}$ components: the evaluated reference simulation ($REF$) and three sensitivity calculations in which the agricultural emissions have been reduced by different percentages, 50% in simulation $REF\_50$, 66% in simulation $REF\_66$ and 100% (i.e., removing all agricultural emissions) in simulation $REF\_100$.





**Table 2.** Summary of the comparison of model data to observations of aerosol component concentrations. OAM and MAM are the spatial arithmetic mean of the observations and the model, respectively (in $\mu g/m^3$). PF2 is the percentage of model results within a factor of two of the observations.

| species | network | MAM | OAM | MAM/OAM | PF2 |
|---------|---------|-----|-----|---------|-----|
| $SO_4^{2-}$ | EPA | 1.22 | 1.18 | 1.03 | 85.5 |
| $SO_4^{2-}$ | EMEP | 1.36 | 1.70 | 0.79 | 86.5 |
| $SO_4^{2-}$ | EANET | 1.54 | 3.30 | 0.46 | 88.8 |
| $NO_3^-$ | EPA | 0.65 | 0.42 | 1.54 | 63.0 |
| $NO_3^-$ | EMEP | 2.08 | 1.15 | 1.81 | 32.6 |
| $NO_3^-$ | EANET | 1.11 | 1.37 | 0.81 | 68.3 |
| $NH_4^+$ | EPA | 0.77 | 0.79 | 0.97 | 88.0 |
| $NH_4^+$ | EMEP | 1.11 | 1.07 | 1.03 | 74.6 |
| $NH_4^+$ | EANET | 0.77 | 0.96 | 0.79 | 80.6 |

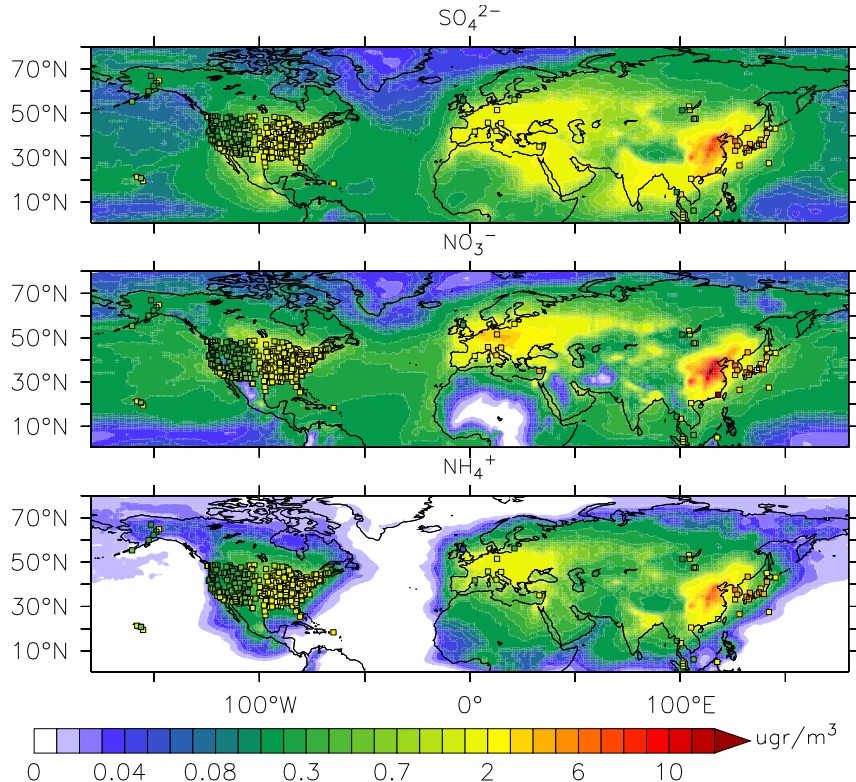

**Figure 4.** Simulated mean concentrations of $PM_{2.5}$ component ($SO_4^{2-}$, $NO_3^-$ and $NH_4^+$) in $\mu g/m^3$ at the surface for the year 2010, with observations from EPA, EMEP and EANET (averaged over the same period) overplotted.



The total primary emitted particle mass from agricultural activities in the $REF$ simulation is 0.4 and 1.9 Tg(C)/yr Black Carbon (BC) and Organic Carbon (OC), respectively, representing in both cases $\sim 5\%$ of their total emissions. The $NO_x$ emissions are 0.7 Tg(N)/yr, i.e., only $\sim 1.7\%$ of the total $NO_x$ emissions. Most importantly, 34.3 Tg(N)/yr of $NH_3$ are emitted by agricultural activities, accounting for $\sim 80\%$ of the anthropogenic and $\sim 67\%$ of the global total ammonia emissions. Finally,

0.1 Tg(S)/yr of $SO_2$ (less than 1% of the total $SO_2$ emissions) and 23.2 Tg(C)/yr of CO ($\sim 5\%$ of the total CO emissions) are emitted, by agricultural waste burning.

Considering these emission magnitudes, the main effects of agricultural emissions on $PM_{2.5}$ are expected from $NH_3$ through gas-particle partitioning. Therefore, the ammonia emissions used in this work have been compared to other used databases, such as EDGARv4.3.1 (Emission Database for Global Atmospheric Research Crippa et al., 2016) and RCP85 (Representative

Concentration Pathways van Vuuren et al., 2011b, a). These datasets differ globally by $\sim 15\%$ (40.26, 47.49 and 40.62 Tg/yr for EDGAR-CIRCE, EDGARv4.3.1 and RCP85, respectively). This reflects the uncertainties in the emission estimates of ammonia which could be up to 50% on a local scale (Beusen et al., 2008). The implementation of bidirectional exchange of ammonia between soil and atmosphere may improve the emissions from livestock, although this approach is still associated with underestimates of emissions (Zhu et al., 2015). Further, ammonia emitted from traffic is included ($\sim 1\%$ of total ammonia

emissions) although toward the lower end of what has been estimated by Sun et al. (2016).

As shown by Lorenz and Steffens (1997); Webb et al. (2006); Kai et al. (2008), a sustainable reduction of ammonia emissions between 20% to 90% could be achieved, depending on the emission process and the methodology applied (e.g., slurry acidification, adjustment in slurry application, under-floor drying of broiler manure in buildings, replace urea with ammonium nitrate). As the efficiencies of the abatement processes are not well established (Misselbrook et al., 2002), fixed relative

reductions have been applied here to all agricultural emissions.

## 3   Results and discussion

### 3.1   Impact on $PM_{2.5}$

In Figure 5 the relative annual average surface $PM_{2.5}$ concentration changes between simulations $REF\_50$, $REF\_66$, $REF\_100$ and $REF$ are presented. These simulations reflect the impact on $PM_{2.5}$ of policies imposing an overall decrease in the agricul-

tural emissions of 50, 66 and 100%, respectively. In Tab. 3 the predicted $PM_{2.5}$ concentrations and pH for all simulations are also listed. The largest effects are found over Europe, North America and over China, the latter with smaller relative intensity. A 50%, 66% and 100% reduction of ammonia emissions would reduce the annual and geographical mean $PM_{2.5}$ levels over Europe by $\sim 1.0\,\mu g/m^3$ (11%), $1.8\,\mu g/m^3$ (19%) and $3.1\,\mu g/m^3$ (34%), respectively, compared to the reference annual surface concentration of $8.9\,\mu g/m^3$. The same relative emission decreases in North America lead to $PM_{2.5}$ concentration reductions of

$0.3\,\mu g/m^3$ (8%), $0.5\,\mu g/m^3$ (12%) and $0.69\,\mu g/m^3$ (16%), respectively, compared to a reference annual surface concentration of $4.0\,\mu g/m^3$. Over East Asia the absolute decrease in the annual average $PM_{2.5}$ concentration near the surface is $1.6\,\mu g/m^3$ (5%), $2.7\,\mu g/m^3$ (8%) and $4.08\,\mu g/m^3$ (13%), respectively, for the three scenarios. Although the absolute changes in East Asia (relative to a reference value of $31.1\,\mu g/m^3$), are larger than the corresponding changes estimated over Europe and North




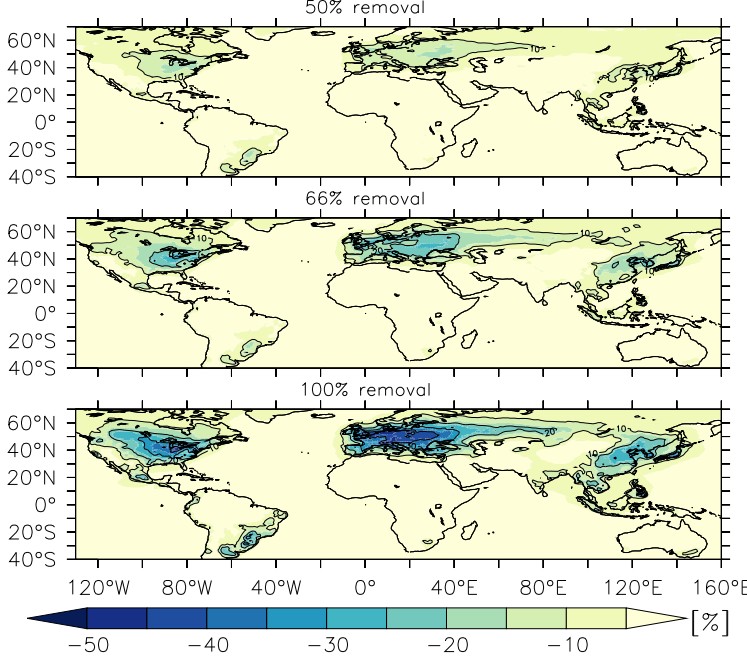

**Figure 5.** Relative annual average surface PM2.5 concentration changes (in %) from the three scenarios with agricultural emissions reductions of 50, 66 and 100% (top, middle and bottom, respectively).

America, the relative changes are smaller. In fact, the fraction of fine particle mass that is directly ammonia sensitive (i.e., $(NH_4^+ + NO_3^-)/PM_{2.5}$) is relatively smaller in East Asia ($\sim$13%) compared to Europe ($\sim$27%) and North America ($\sim$17%), and reduction of $NH_3$ emissions would mainly decrease the nitrate and ammonium components, rather than the predominant components of $PM_{2.5}$ in this part of the world. Over South Asia, this effect is even more enhanced. The decreased emissions, in fact, have a negligible impact on annual average $PM_{2.5}$, reducing it by 0.62 (2%), 0.76 (3%) and 1.44 (6%) $\mu g/m^3$, for reductions of ammonia emissions of 50%, 66% and 100%, respectively. The fraction of fine particle mass sensitive to ammonia, in this region, is very low (3%), since more than 90% of the aerosol mass is not formed by the ammonium-sulfate-nitrate components, but rather by organic carbon ($\sim$ 45 % of the total mass) and dust ($\sim$ 35% of the total mass).

In all four regions considered here the impact of $NH_3$ emissions reduction on $PM_{2.5}$ concentrations is strongest during winter. This is related to the enhanced $NH_4NO_3$ partitioning in the gas phase due to the higher temperatures during summer, so that a reduction of $NH_3$ influences the gas phase concentrations more strongly than the particulate phase during this season. The opposite happens during the winter season. Additionally, in the $REF$ simulation, the winter total nitrate (gas and aerosol) concentrations are somewhat higher than during the summer over Europe (5.3 vs 4.5 $\mu g/m^3$), USA (1.5 vs 1.0 $\mu g/m^3$), South Asia (10.0 vs 3.4) and East Asia (8.2 vs 4.5 $\mu g/m^3$). This is related to the lower boundary layer height in winter, and hence less dilution of the emitted tracers. In addition, the total emissions in winter are higher than the emissions in summer. As extensively discussed in Pozzer et al. (2012a), this is due to a dual peak present in the emissions database, which leads to increased $NH_3$





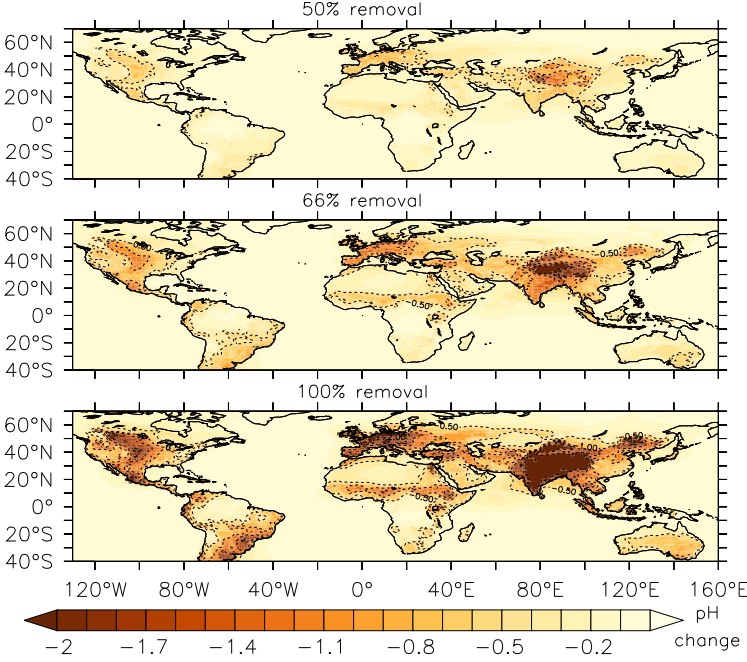

**Figure 6.** Absolute annual average surface aerosol pH changes (all modes) from three scenarios with agricultural emission reductions of 50, 66 and 100% (top, middle and bottom, respectively).

emissions in late winter/ beginning of spring and late summer/beginning of autumn, which most probably does not realistically reproduce the real seasonality of the ammonia emissions (Paulot et al., 2014).

The total $PM_{2.5}$ sulfate (i.e., $SO_4^{2-} + HSO_4^-$) is not directly affected by $NH_3$ emission reductions since it can exist in the aerosol phase in the form of ammonium sulfate or ammonium bisulfate, depending on the ammonium concentration.

5 However, sulfate formation in the aqueous phase is limited by high acidity. As a consequence, the $SO_4^{2-}$ concentration in $PM_{2.5}$ decreases, annually averaged, by 11%, 23%, and 66% over Europe, by 15%, 28% and 57% over North America, by 3%, 7% and 50% over South Asia, and by 18%, 36% and 74% over East Asia for a reduction of 50%, 66% and 100% of agricultural emissions, respectively. This is counterbalanced by an increase of $HSO_4^-$ concentrations.

For Europe and North America, the simultaneous decrease of nitrate and ammonium, makes the reduction of agricultural
10 emissions very effective, especially during winter, in accord with the findings of Tsimpidi et al. (2007) and Megaritis et al. (2013). Furthermore, the relationship between ammonia and $PM_{2.5}$ concentrations is not linear, and is governed by the sulfate/nitrate ratio (Tsimpidi et al., 2007). Our EMAC simulations reveal that the $PM_{2.5}$ response to $NH_3$ emissions is more linear during winter (compared to summer) since the sulfate/nitrate ratio is generally lower.

Following Makar et al. (2009), the particle neutralization ratio (PNR, i.e. $(NH4^+)/(2(SO_4^{2-} + HSO_4^-) + NO3^-)$) calcula-
15 tions indicate that in Europe and East Asia (both with PNR equal to 0.20) ammonia concentrations must be decreased relatively more strongly than in North America and South Asia (PNR equal to 0.13 for both regions) to reach the ammonia limited





regime, i.e., before $PM_{2.5}$ can be efficiently controlled by decreasing $NH_3$ emissions. On the other hand, the absolute reduction in $PM_{2.5}$ depends on the fraction of fine particulate mass that is directly ammonia sensitive. As a consequence, Europe has the overall largest potential of reducing annual averaged $PM_{2.5}$ by strongly controlling $NH_3$ emissions (up to 34%), followed by North America (up to 16%) and East Asia (up to 13%), while South Asia has very limited potential (up to 6%). Thus it follows

that, although the emission decrease needed in Europe to reach the ammonia limited regime is larger than in North America, the effective gain of further reduction - once this regime is reached - is considerably larger. In East Asia, where $PM_{2.5}$ is not ammonia limited, even strong emission decreases would reduce the $PM_{2.5}$ mass by up to to 13% on annual average.

## 3.2 Impact on particle pH

In addition to the significant reductions in $PM_{2.5}$ from ammonia emission controls, which are considered beneficial for human

health, we note that the aerosol pH can change substantially. This has the potential of altering the particle liquid phase and heterogeneous chemistry, including reactive uptake coefficients and the outgassing of relatively weak acids, and the pH of cloud droplets that grow on aerosols, which in turn affects aqueous phase sulfate formation. Ammonia is in fact the most abundant and efficient base that controls the aerosol composition over anthropogenically influenced regions, which neutralizes sulfuric, nitric and other acids.

In the $REF$ simulation, the particles over the focal regions are highly acidic, consisting mainly of ammonium sulfate and ammonium nitrate, as also shown by Weber et al. (2016). Figure  6 illustrates how the aerosol pH can drop due to $NH_3$ emission decreases. Over Europe, the calculated mean pH decreases of aerosols are 0.35, 0.62 and 1.05 pH units for the $REF\_50$, $REF\_66$ and $REF\_100$ simulations, respectively. The calculations indicate similar decreases over East Asia (0.35, 0.62 and 1.11 pH units, respectively), smaller decreases over North America (0.16 ,0.29 and 0.51 pH units, respectively), while the

largest decreases are present over South Asia (0.56, 0.99 and 1.72, pH units, respectively). Over South Asia, the impact of ammonia emissions reduction on pH is the largest (see Fig. 6) despite the relatively small impact of the same changes on $PM_{2.5}$. This is due to the high sulfate concentrations, which are neutralized in decreasing order by the presence of ammonium in the three sensitivity simulations. The pH of $PM_{2.5}$ is therefore more sensitive to ammonia emissions (and its atmospheric concentrations) than sulfate, as shown by Weber et al. (2016). This increase of acidity for reduced ammonia emissions would

have strong influence on halogen activation and aerosol-gas equilibrium of weak acids in the atmosphere.

Contrary to what was found for $PM_{2.5}$, the reduction of pH is larger during summer than during winter. This is due to the lower concentrations of ammonia in the aerosol phase during summer (see Sect. 3.1), i.e., with relatively low neutralization capability in this season. Therefore, any further reduction of ammonia emissions would strongly reduce the neutralization potential, and therefore increase even more drastically the acidity of the particles.

It should be mentioned that in the present calculations the chemical impact of alkaline desert dust is not taken into account, which can contribute significantly to $PM_{2.5}$ over areas downwind of the deserts (Karydis et al., 2016), e.g., over the Indian subcontinent in the dry season and over East China during spring (Wang et al., 2013), so that the pH effect described here is probably an upper limit. This topic is subject of a publication in preparation.



**Table 3.** Average concentration of $PM_{2.5}$ and $PM_{2.5}$ components (in $\mu g/m^3$). $SO_4^{2-}$ represent total sulfate (i.e. $SO_4^{2-}$ and $HSO_4^-$). pH average values are also listed.

| Region | REF simulation | | | | | REF_50 simulation | | | | | REF_66 simulation | | | | | REF_100 simulation | | | | |
|---|---|---|---|---|---|---|---|---|---|---|---|---|---|---|---|---|---|---|---|---|
| | $NH_4^+$ | $NO_3^-$ | $SO_4^{2-}$ | $PM_{2.5}$ | pH | $NH_4^+$ | $NO_3^-$ | $SO_4^{2-}$ | $PM_{2.5}$ | pH | $NH_4^+$ | $NO_3^-$ | $SO_4^{2-}$ | $PM_{2.5}$ | pH | $NH_4^+$ | $NO_3^-$ | $SO_4^{2-}$ | $PM_{2.5}$ | pH |
| **All year** | | | | | | | | | | | | | | | | | | | | |
| Europe | 0.94 | 1.80 | 1.25 | 8.95 | 2.04 | 0.72 | 1.32 | 1.20 | 7.93 | 1.68 | 0.53 | 0.92 | 1.19 | 7.22 | 1.42 | 0.09 | 0.27 | 1.19 | 5.89 | 0.98 |
| North America | 0.27 | 0.45 | 0.56 | 4.07 | 1.60 | 0.20 | 0.30 | 0.55 | 3.73 | 1.43 | 0.15 | 0.21 | 0.54 | 3.58 | 1.31 | 0.06 | 0.11 | 0.54 | 3.38 | 1.09 |
| South Asia | 0.50 | 0.39 | 1.41 | 23.27 | 2.87 | 0.46 | 0.25 | 1.41 | 22.65 | 2.31 | 0.42 | 0.18 | 1.41 | 22.51 | 1.88 | 0.16 | 0.12 | 1.40 | 21.83 | 1.15 |
| East Asia | 1.56 | 2.43 | 2.51 | 31.12 | 1.95 | 1.12 | 1.47 | 2.49 | 29.50 | 1.59 | 0.77 | 0.80 | 2.49 | 28.43 | 1.33 | 0.14 | 0.10 | 2.49 | 27.04 | 0.83 |
| World | 0.10 | 0.21 | 0.32 | 9.23 | 1.84 | 0.08 | 0.16 | 0.32 | 9.05 | 1.75 | 0.06 | 0.13 | 0.32 | 8.98 | 1.68 | 0.02 | 0.10 | 0.32 | 8.89 | 1.53 |
| **Summer** | | | | | | | | | | | | | | | | | | | | |
| Europe | 0.90 | 1.02 | 1.89 | 7.74 | 2.26 | 0.74 | 0.70 | 1.88 | 7.07 | 1.80 | 0.60 | 0.45 | 1.91 | 6.72 | 1.50 | 0.11 | 0.10 | 1.87 | 5.70 | 1.04 |
| North America | 0.22 | 0.13 | 0.68 | 5.51 | 1.93 | 0.18 | 0.08 | 0.68 | 5.32 | 1.73 | 0.14 | 0.06 | 0.66 | 5.29 | 1.59 | 0.06 | 0.05 | 0.67 | 5.23 | 1.34 |
| South Asia | 0.17 | 0.19 | 0.75 | 16.76 | 2.96 | 0.16 | 0.18 | 0.74 | 16.49 | 2.44 | 0.14 | 0.17 | 0.74 | 16.44 | 2.07 | 0.06 | 0.16 | 0.73 | 16.16 | 1.40 |
| East Asia | 1.21 | 0.98 | 3.00 | 19.33 | 1.87 | 0.89 | 0.50 | 2.98 | 18.40 | 1.57 | 0.61 | 0.19 | 2.94 | 17.69 | 1.36 | 0.09 | 0.02 | 2.93 | 17.04 | 0.95 |
| World | 0.09 | 0.13 | 0.36 | 7.39 | 1.94 | 0.07 | 0.11 | 0.36 | 7.25 | 1.85 | 0.05 | 0.09 | 0.36 | 7.23 | 1.78 | 0.01 | 0.08 | 0.36 | 7.20 | 1.64 |
| **Winter** | | | | | | | | | | | | | | | | | | | | |
| Europe | 1.08 | 2.48 | 0.80 | 11.12 | 1.90 | 0.80 | 1.96 | 0.75 | 9.84 | 1.59 | 0.55 | 1.46 | 0.74 | 8.86 | 1.35 | 0.06 | 0.47 | 0.74 | 6.94 | 0.90 |
| North America | 0.43 | 1.01 | 0.48 | 3.98 | 1.39 | 0.29 | 0.68 | 0.45 | 3.36 | 1.22 | 0.20 | 0.47 | 0.44 | 2.98 | 1.10 | 0.06 | 0.19 | 0.43 | 2.48 | 0.87 |
| South Asia | 0.71 | 0.57 | 1.75 | 29.63 | 2.95 | 0.64 | 0.33 | 1.75 | 28.65 | 2.40 | 0.58 | 0.20 | 1.75 | 28.48 | 1.90 | 0.24 | 0.11 | 1.75 | 27.54 | 1.03 |
| East Asia | 2.07 | 4.25 | 2.00 | 40.16 | 2.18 | 1.53 | 2.96 | 1.91 | 37.96 | 1.73 | 1.06 | 1.84 | 1.90 | 36.27 | 1.39 | 0.18 | 0.21 | 1.93 | 33.61 | 0.72 |
| World | 0.13 | 0.33 | 0.30 | 11.39 | 1.78 | 0.10 | 0.25 | 0.29 | 11.14 | 1.68 | 0.07 | 0.19 | 0.29 | 11.02 | 1.60 | 0.02 | 0.12 | 0.29 | 10.85 | 1.43 |



### 3.3 Impact on public health

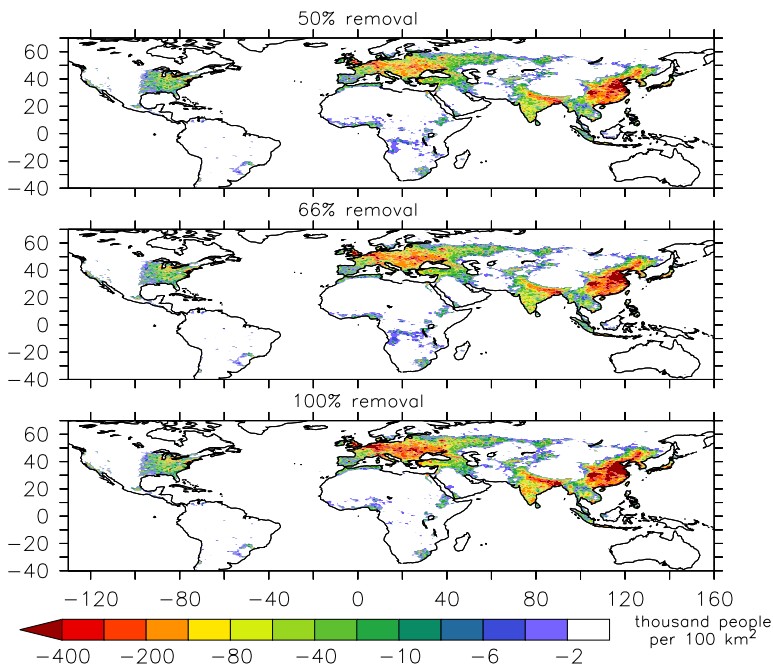

**Figure 7.** Annual average mortality attributable to $PM_{2.5}$ concentration changes (in $\mathrm{thousands\,people}/100\,\mathrm{km^2}$) from the three scenarios with agricultural emissions reductions of 50, 66 and 100% (top, middle and bottom, respectively).

From the simulated $PM_{2.5}$ concentrations, the mortality attributable to air pollution has been calculated following the method of Lim et al. (2013) using the exposure-response functions of Burnett et al. (2014) which shows how fine particulate matter is associated with health impacts, through chronic obstructive pulmonary disease (COPD), acute lower respiratory infections (ALRI), cerebrovascular disease (CEV), ischaemic heart disease (IHD), and lung cancer (LC). Here mortality attributable to $PM_{2.5}$ at 50% relative humidity has been estimated, thus not accounting for ozone related mortality through COPD, which is $\sim 5\%$ of the total mortality attributable to air pollution (Lelieveld et al., 2015). The results, presented in Tab. 4 and Fig.7, show that a reduction of 50% in agricultural emissions could have a large impact on air pollution related mortality, reducing it worldwide by $\sim 8\%$ i.e., affecting 250 (95% confidence interval (CI): 148-290) thousand people/year. North America would benefit from a large relative change, reducing the number of deaths by $\sim 30\%$ (16 (95% CI: 10-19) thousand people/year), followed by Europe ($\sim 19\%$, 52 (95%CI: 41-53) thousand people/year) and East Asia ($\sim 8\%$, 105 (95% CI: 53-116) thousand people/year), while almost no effects are found over South Asia ($\sim 3\%$, 25 (95% CI: 14-33) thousand people/year). The relatively large effect in North America is explained by the shape of the integrated response function (Burnett et al., 2014), which predicts a steep change in the attributable fraction at relative low $PM_{2.5}$ concentrations. If it were possible to fully phase out agricultural emissions, the global reduction of $PM_{2.5}$ related mortality would reduce by about 801 thousand people per year (95% CI: 417-984). In Europe the number would be reduced by about 222 thousand (95% CI: 139-249), in North America





by 40 thousand (95% CI: 17-61), in East Asia by about 343 thousand per year (95% CI: 159-401) and in South Asia by 82 thousand per year (95% CI: 45-110) (Table 4).

Ammonia reduction policies should consider the intricate and non-linear effects through gas-aerosol partitioning and multi-phase chemistry (including aerosol pH changes) and therefore a coherent decrease of ammonia, nitrogen and sulfur emissions is

recommended. A coupled reduction of $NH_3$ and acid precursor emissions (e.g., $SO_2$) cannot only limit the decrease in aerosol pH but can also lead to a more efficient reduction of $PM_{2.5}$ concentrations than the $NH_3$ emission control alone, as shown by (Tsimpidi et al., 2007). In the electronic supplement, a table showing the changes in mortality for the top 100 most populated countries is presented. Consistently with the results of Lee et al. (2015), Central and East European countries would benefit strongly from agricultural emission reductions , decreasing drastically the per capita air pollution related mortality. This can

be seen also in Fig. 5, as the strongest relative changes in $PM_{2.5}$ due to agricultural emissions reduction are found in Central and East Europe, where a 50% emission reduction would decrease mortality attributable to air pollution by $\sim$ 15-20%.

It must be emphasized that, although many epidemiological studies have linked long term $PM_{2.5}$ exposure to public health outcome, it is yet unclear if any particular aerosol components and/or source categories are predominantly responsible for air pollution related mortality. The debate is open and firm conclusions of a specific relationship have not been reached (Harrison

and Yin, 2000; Reiss et al., 2007), although it is expected that some aerosol components may be more toxic than others (Shiraiwa et al., 2012).

**Table 4.** Mortality attributable to air pollution in thousands people per year. In parenthesis the minimum-maximum range.

| Region | REF | | REF_50 | | REF_66 | | REF_100 | |
|---|---|---|---|---|---|---|---|---|
| | average | range | average | range | average | range | average | range |
| Europe | 277 | ( 148 - 414) | 225 | ( 107 - 361) | 176 | ( 66 - 313) | 55 | ( 9 - 165) |
| North America | 54 | ( 21 - 100) | 38 | ( 11 - 81) | 26 | ( 6 - 65) | 14 | ( 4 - 39) |
| South Asia | 778 | ( 410 - 1140) | 753 | ( 396 - 1107) | 750 | ( 395 - 1102) | 696 | ( 365 - 1030) |
| East Asia | 1381 | ( 607 - 1929) | 1275 | ( 553 - 1812) | 1195 | ( 514 - 1719) | 1037 | ( 447 - 1527) |
| World | 3155 | (1523 - 4603) | 2905 | (1375 - 4313) | 2739 | (1280 - 4123) | 2353 | (1106 - 3619) |

## 4 Conclusions

Pinder et al. (2007) showed that in North America emission controls of $SO_2$ and $NO_x$ are likely to be very costly and probably less efficient than decreasing agricultural emissions. Therefore, the regulation of ammonia emissions from agricultural activities

offers the possibility of relatively cost-effective control policies for $PM_{2.5}$. Our model simulations indicate that a 50% decrease of ammonia emissions could reduce the annual, geographical average near-surface $PM_{2.5}$ concentrations up to $\sim$ 1.0 (11%), 0.3 (8%), 1.6 (5%) and 0.6 (2%) $\mu g/m^3$ in Europe, North America, East Asia and South Asia, respectively. The reduction can even be larger during winter (up to $\sim$ 1.3 (11%), 0.6 (15%), 2.2 (5%) and 1.0 (3%) $\mu g/m^3$, respectively) when particulate ammonium nitrate concentrations are typically higher than in summer.





Our model simulations underscore that strong non-linearity plays a role in the sulfate-nitrate-ammonia system, which affects the efficiency of $PM_{2.5}$ controls, especially during summer when the sulfate/nitrate ratio is high. A strong reduction of $PM_{2.5}$ in response to $NH_3$ emission regulation is expected once the ammonia-limited regime is reached. As a result, the possible $PM_{2.5}$ reduction could be as large as ~34% and ~17% in Europe and North America, respectively. Our results also suggest that ammonia emission controls could reduce the particle pH up to 1.5 pH units in East Asia during winter, and more than 1.7 pH units in South Asia, theoretically assuming complete agricultural emission removal, which could have repercussions for the reactive uptake of gases from the gas phase and the outgassing of relative weak acids.

Furthermore, the global mortality attributable to $PM_{2.5}$ could be reduced by ~ 250 (95% CL: 148-290) thousand people/year worldwide by decreasing agricultural emissions by 50%, with a gain of 16 (30%), 52 (19%), 105 (8%) and 25 (3%) thousand people/year in Europe, North America South Asia and East Asia, respectively. A total phase-out of agricultural emissions would even reduce the mortality attributable to air pollution worldwide by about 801 (25%) thousand people/year, in Europe by about 222 (80%) thousand people/year, in North America by about 40 (74%) thousand people/year, in South Asia by about 82 (10%) thousand people/year and in East Asia by about 343 (25%) thousand people/year. These strong impacts are related to the non-linear responses in both the sulfate-nitrate-ammonia system and the exposure-response functions at relatively low $PM_{2.5}$ concentrations.

Therefore, emission control policies, especially in North America and Europe, should involve strong ammonia emission decreases to optimally reduce $PM_{2.5}$ concentrations.

*Data availability.* The data from all model integrations are available from the authors upon request.

*Competing interests.* No competing interests are present.

*Acknowledgements.* V. Karydis acknowledges support from a FP7 Marie Curie Career Integration Grant (project reference 618349). A. P. Tsimpidi acknowledges support from a DFG Individual Grant Programme (project reference TS 335/2-1).





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
