# Peer review of "Impact of agricultural emission reductions on fine particulate matter and public health."

_Atmospheric Chemistry and Physics, 2017_

## Referee Comment (RC1) · Anonymous Referee #1 · 16 Jul 2017

An interesting an important topic discussed so far mostly at a regional level. Presenting the global perspective is interesting but I personally doubt that such work has any implications on regional policy as it entirely misses discussion of regionally specific aspect of mitigation opportunities analyzing rather unrealistic scenarios of agricultural emissions; additionally referring to 2010 levels while emissions from livestock and arable land production (fertilizer use) are likely to increase further in several regions, especially in Asia. The above does not disqualify the paper and the authors do not claim that this analysis could guide regional policy but I believe that more discussion, or at least a clear statement, of this aspect is needed. Another aspect of this work that needs more clarity is the issue of temporal distribution of agricultural emissions used in the simulations. The authors make comments about the potential issue with temporal

pattern of emissions but do not explain that any further and do not provide the actually used profile which makes it difficult to comment on that further. Another issue is the spatial resolution of the modelled PM2.5 concentrations and its use for calculation of population exposure. While the dose-response functions are referred to, it is not clear how the 10x10km (or 0.1x0.1 degree) mortality map (Fig 7) is produced when the output of the model is 1.1x1.1 degree which would lead to underestimation of exposure in urban areas. A clear explanation and discussion of consequences for the results and conclusions would be important.

More specific comments: ABSTRACT: I am not sure if the last sentence about the impact of 100% reduction is of any significance; such reductions are not even theoretically possible. INTRODUCTION: Page 1, Line 23: One could add there a reference to the EU policy which includes now targets for NH3 emissions within the revised air quality legislation. The authors include a reference to that later in the paper. Page 2, Line 4: not clear what is meant by 'manure processing' , suggest replacing with 'manure storage and on field application' Page 2, line 4: suggest add 'N' or 'nitrogen' before "fertilizer" Page 2, line 12: maybe 'leads' should be replaced with 'would lead' or 'could lead' as this is a modelling study rather than impact observed anywhere. Page 2, line 18: 'by agriculture' should be replaced with 'from agriculture' and 'resulted' can be possibly modified to 'would or could result' Page 2, last paragraph: As before, suggest adding a reference to the recent European air quality policy and possibly underlying analysis. METHODOLOGY Page 3, from line 21: The emissions are for the year 2010 but the references are for data sets until 2005. Few words of explanation? Page 4, Figure 2: A bit small, hard to read the axis Page 5, last paragraph: I believe it would be beneficial to put these assumptions in perspective of what has been discussed as feasible since the reductions given here, even the lowest level, are in most regions perceived as either infeasible or close to maximum reduction potential unless dietary changes are considered reducing meat demand. Beyond that, the realistic potential varies strongly between the regions which could be at least mentioned. It would be also advisable to add a clear statement

which agricultural sources are included, eg., livestock manure, N mineral fertilizers, open burning of agricultural residues. . .. Page 6; first paragraph: Presumably the first sentence refers to agricultural burning and so it could be moved to the end of this paragraph where combustion emissions are mentioned. In general this paragraph should be clear as to which sources are meant next to specific pollutants. Page 6; line 17: The 20-90% reduction refers to single measures and not to the overall mitigation potential and so nowhere 90% can be achieved for the whole agriculture. The potential is typically between 20-45% with some exceptions where structure is different, i.e., China with large share of emissions from urea and ammonium bicarbonate. Discussion of mitigation potential is available for a some countries, e.g., the Netherlands, Denmark, UK but also European studies discuss this; for example: https://link.springer.com/chapter/10.1007%2F978-94-017-9722-1_9 and specific discussion of feasibility, experience, and obstacles in reducing emissions in the UNECE: http://www.unece.org/env/documents/2007/eb/wg5/WGSR40/ece.eb.air.wg.5.2007.13.e.pdf RESULTS Page 8, para from line 15: Total emissions in winter are not higher than in summer! NH3 emissions are increasing with temperature and also organic fertilizers are applied in Spring, Summer, Autumn, just as the mineral fertilizers. I have mentioned the issue of temporal distribution of emissions earlier; I think if the distribution is indeed wrong then the consequences of that for the simulation results need to be discussed more thoroughly. Figure 7; here the resolution for the mortality attributable to PM2.5 is indicated as 10x10km. An explanation what data are used to develop that is needed. In general some discussion related to how coarse resolution concentration fields are used in health impact assessment would be useful, see eg. http://www.sciencedirect.com/science/article/pii/S1364815215000808?via%3Dihub

---

## Referee Comment (RC2) · Anonymous Referee #2 · 18 Jul 2017

I recommend this manuscript be published after minor reversions. (1) Please state if the ammonia reductions in this study are feasible, especially for different regions. (2) Section 3.2, the aerosol pH would be determined by aerosol water, which also depends on the secondary nitrate and sulfate concentrations, relative humidity etc. Further, rich or poor ammonia in different regions should have markedly different effects on aerosol pH. Please have some discussions on them. (3) The epidemiological studies did find the secondary inorganic aerosols could have negligible effects on human health, it is better the authors reword it, Line 12, Page13. The references: Mar TF et al., PM source apportionment and health effects. 3. Investigation of inter-method variations in associations between estimated source contributions of PM2.5 and daily mortality in Phoenix, AZ. JOURNAL OF EXPOSURE SCIENCE AND ENVIRONMENTAL EPIDEMIOLOGY

2006(16), 311–320.; Ito K et al. PM source apportionment and health effects: 2. An investigation of intermethod variability in associations between source-apportioned fine particle mass and daily mortality in Washington, DC. JOURNAL OF EXPOSURE SCIENCE AND ENVIRONMENTAL EPIDEMIOLOGY 2006(16), 300—310).

---

## Author Comment (AC1) · 15 Aug 2017

We thank the referee #1 for the accurate review.

Main comments:

1. **Presenting the global perspective is interesting but I personally doubt that such work has any implications on regional policy as it entirely misses discussion of regionally specific aspect of mitigation opportunities analyzing rather unrealistic scenarios of agricultural emissions; additionally referring to 2010 levels while emissions from livestock and arable land production (fertilizer use) are likely to increase further in several regions, especially in Asia.**

[Figure]

This study presents the effect on atmospheric chemistry of hypothetical reductions of agricultural emissions. Although these reductions are hard to reach in reality, the model results allow us to understand the link between ammonia and $PM_{2.5}$ for different ammonia levels. Even though a 100% reduction of ammonia emissions cannot be achieved by policies, it is of scientific interest to study the impacts of different reduction scenarios. Further, the experiments show that, although a reduction of ammonia would indeed reduce the fine particle concentrations, the particle acidity would be strongly affected, and these effects must be taken in account when establishing emission reduction policies. In the manuscript we have listed a few of the numerous studies which analyzed technologies for ammonia emissions reduction from livestock production or manure/slurry application. A good overview is given by Webb et al. (2006), who reviewed different methodologies for ammonia emissions reduction. Webb et al. (2006) showed that different technologies could produce different emissions reduction, ranging from 20% to 80%. In the same study it is shown that for the United Kingdom a moderate reduction in ammonia emission is easily affordable, while the costs are likely to increase exponentially for reductions above 25%. The same control measures would become even more difficult to apply in developing countries or where strong ammonia emissions are present (such as South Asia), as these should be adopted massively. Further, as noticed by the referee, livestock production is projected to largely increase in the Asia region (Delgado et al., 2001). On the other side, the societal cost for hospitalization and/or premature mortality due to air pollution should be also considered, as the large expenses for ammonia emissions reduction could be compensate by a reduction in premature mortality. This is however beyond the present study and a further work in this direction is planned.

2. **Another aspect of this work that needs more clarity is the issue of temporal distribution of agricultural emissions used in the simulations. The authors**

**make comments about the potential issue with temporal pattern of emissions but do not explain that any further and do not provide the actually used profile which makes it difficult to comment on that further.**

We thank the referee for pointing this out. Indeed the description in the manuscript was not accurate enough. As shown by Pozzer et al. (2012) the comparison between observed and modelled $NH_4^+$ shows a good temporal distribution of ammonia emissions both for Europe and Asia (temporal correlations are above 0.7 and 0.5, respectively). On the other hand, the correlation on the East Coast of the USA is even below $0.2$. As suggested by Pozzer et al. (2012), this is possibly due to the wrong seasonality, driven by an understimation of the livestock emissions, which have a maximum in summer and should account for 80% of the annual $NH_3$ emissions in the region (Battye et al., 2003). Differently, the agricultural emissions of ammonia in this region in the model reproduce mostly the fertilizer application as described by Goebes et al. (2003). We can therefore assume, based on the previous evaluation of the model, that this inaccurate seasonality does affect only the USA region. On the other side, despite the low seasonal correlation with the observations, it should still be mentioned that the overall results over the USA have good agreement with the observations, as the annual mean is well reproduced by the model (see Tab. 2 in the manuscript). We will extend the description in the model, mentioning this issue and that the seasonal results over the USA should be taken with caution.

3. **another issue is the spatial resolution of the modelled PM2.5 concentrations and its use for calculation of population exposure. While the dose-response functions are referred to, it is not clear how the 10x10km (or 0.1x0.1 degree) mortality map (Fig 7) is produced when the output of the model is 1.1x1.1 degree which would lead to underestimation of exposure in urban areas. A clear explanation and discussion of consequences for the results and conclusions would be important.**

We do agree with the referee that this point was not clear enough in the manuscript. The approached used in this manuscript is described in Lelieveld et al. (2015), where a detailed discussion on the uncertainties is included. Although it is expected to have an underestimation of exposure in urban areas, it is also true that model results with such resolution gives very similar outcome (for mortality) when compared to high resolution data as the one used by the Global Burden of Disease (Lim et al., 2013), indicating that the error introduced by the coarse resolution is only marginal. As discussed in the supplement of Lelieveld et al. (2015), $PM_{2.5}$ concentrations are not very sensitive to local emissions as the main $PM_{2.5}$ fraction is secondary in nature. This leads to small urban increments, as shown in many studies that have compared urban with sub-urban and rural $PM_{2.5}$ concentrations. This is particularly true for annual average concentrations, which are most relevant for long-term health impacts and mortality. Further, it has been shown by McKeen et al. (2007), in an intercomparison model exercise, that increasing the resolution of the model could degrade the correlation between model and observed fine particle and that the physic/chemistry package of the model is as important as the resolution. Additionally, a study by de Meij et al. (2007) showed that the mass fractions of $SO_4^{2-}$, $NH_4^+$, $NO_3^-$ are not strongly resolution dependent. Finally, the comparison of the model with observations showed an agreement similar to regional model (see Table.1 and 2) and therefore we assume that the accuracy of our model is comparable to high resolution models. We acknowledge however the lack of discussion on this point in the manuscript and the text will be extended to include also a description of uncertainties sources. Following the work of Lelieveld et al. (2015), an uncertainties of 50% should be estimated for the mortality attributable to air pollution.

Minor comments:

**ABSTRACT: I am not sure if the last sentence about the impact of 100% reduc-**

**tion is of any significance; such reductions are not even theoretically possible.**
We do fully agree with the referee. However, this is a theoretical study, and it is scientifically interesting to check the effect on aerosol acidity for such large decrease of agricultural emissions. Although this reduction is not achievable, we believe to be an important exercise to understand the effect of ammonia on the chemical properties of the atmosphere.

**Page 1, Line 23: One could add there a reference to the EU policy which includes now targets for $NH_3$ emissions within the revised air quality legislation. The authors include a reference to that later in the paper.**
We will move the reference to this location.

**Page 2, Line 4: not clear what is meant by 'manure processing' , suggest replacing with 'manure storage and on field application'**
We will reformulate the sentence.

**Page 2, line 4: suggest add 'N' or 'nitrogen' before "fertilizer"**
The change will be implemented.

**Page 2, line 12: maybe 'leads' should be replaced with 'would lead' or 'could lead' as this is a modelling study rather than impact observed anywhere.**
We agree with the referee and the change will be implemented

**Page 2, line 18: 'by agriculture' should be replaced with 'from agriculture' and 'resulted' can be possibly modified to 'would or could result'**
The changes will be implemented.

**Page 2, last paragraph: As before, suggest adding a reference to the recent European air quality policy and possibly underlying analysis.**

A good review of air quality policy both for Europe and US is presented in Kuklinska et al. (2015). We will refer to this publication in the revised manuscript, as well mentioning the official web site reviewing the European air quality policy.

**Page 3, from line 21: The emissions are for the year 2010 but the references are for data sets until 2005. Few words of explanation?**
The references are indeed based on the year 2005, although they are for the same model set-up and emissions database. However, an additional evaluation is present in the manuscript, both with $PM_{2.5}$ (climatology) and with aerosol inorganic components (for the year 2010). This corroborates that the model performances have not been deteriorated with respect to the previous evaluation when simulating a different year.

**Page 4, Figure 2: A bit small, hard to read the axis**
The axis label will be increased.

**Page 5, last paragraph: I believe it would be beneficial to put these assumptions in perspective of what has been discussed as feasible since the reductions given here, even the lowest level, are in most regions perceived as either infeasible or close to maximum reduction potential unless dietary changes are considered reducing meat demand. Beyond that, the realistic potential varies strongly between the regions which could be at least mentioned. It would be also advisable to add a clear statement which agricultural sources are included, eg., livestock manure, N mineral fertilizers, open burning of agricultural residues.**
As pointed out also by referee #2, a more detailed discussion on how feasible such reductions are is needed in the manuscript. We will extend the text at page 7, lines 16-20. In fact, although the technology to reduce ammonia from livestock production (the largest source of European anthropogenic emissions of ammonia) by 80% does exist, it is unclear about the costs of such abatement methods.

Webb et al. (2006) suggested that "[...]While there is scope for reduction in $NH_3$ emissions at moderate cost, to achieve large ($> 25\%$) reductions costs are likely to increase exponentially."

**Page 6; first paragraph: Presumably the first sentence refers to agricultural burning and so it could be moved to the end of this paragraph where combustion emissions are mentioned. In general this paragraph should be clear as to which sources are meant next to specific pollutants.**
We will reformulate the text to make it clearer, moving the emissions of Black Carbon and Organic Carbon at the end of the paragraph.

**Page 6; line 17: The 20-90% reduction refers to single measures and not to the overall mitigation potential and so nowhere 90% can be achieved for the whole agriculture. The potential is typically between 20-45% with some exceptions where structure is different, [..]**
We fully agree with the referee and this chapter will be extended based on the previous comments (see also reply to referee #2).

**Page 8, para from line 15: Total emissions in winter are not higher than in summer! $NH_3$ emissions are increasing with temperature and also organic fertilizers are applied in Spring, Summer, Autumn, just as the mineral fertilizers.**
The referee is correct, and the text was simply wrong. The average emissions in the Northern Hemisphere show clearly a minimum during winter and a maximum in the spring season, with high emissions until late autumn. Here we were referring to the emissions in the USA, which do not present a correct seasonality. As showed by Pozzer et al. (2012), the temporal correlation between model results and observations over the USA for $NH_4^+$ is not fully satisfactory. The emissions of $NH_3$ in this region probably represent fertilizer application (Goebes et al., 2003), underestimating the importance of livestock emissions which have a maximum in summer and should account for 80% of the NH3 emissions in the region (Battye

et al., 2003). This point will be clarified in the text.

**Figure 7: here the resolution for the mortality attributable to $PM_{2.5}$ is indicated as 10x10km. An explanation what data are used to develop that is needed. In general some discussion related to how coarse resolution concentration fields are used in health impact assessment would be useful [. . .]**

The data at $1 \times 1$ degree resolution have been interpolated to the $0.1 \times 0.1$ degree resolution of the population map. As suggested by the referee earlier, this could cause smoothing of the $PM_{2.5}$ values, underestimating the exposure over urban areas. Although this is in principle correct, we have shown in Lelieveld et al. (2015) that this approach gives very similar results to other approaches using very high resolution datasets for $PM_{2.5}$. We therefore consider our results robust in the statistical sense, although an uncertainties of 50% must be considered for our results, in light of such assumptions.

**References**

Battye, W., Aneja, V. P., and Roelle, P. A.: Evaluation and improvement of ammonia emissions inventories, Atmos. Environ., 37, 3873 – 3883, doi:DOI:10.1016/S1352-2310(03)00343-1, 2003.

de Meij, A., Wagner, S., Cuvelier, C., Dentener, F., Gobron, N., Thunis, P., and Schaap, M.: Model evaluation and scale issues in chemical and optical aerosol properties over the greater Milan area (Italy), for June 2001, Atmospheric Research, 85, 243–267, doi:10.1016/j.atmosres.2007.02.001, 2007.

Delgado, C., Rosegrant, M., Steinfeld, H., Ehui, S., and Courbois, C.: Livestock to 2020: the next food revolution, Outlook on Agriculture, 30, 27–29, 2001.

Goebes, M. D., Strader, R., and Davidson, C.: An ammonia emission inventory for fertilizer application in the United States, Atmos. Environ., 37, 2539 – 2550, doi:DOI:10.1016/S1352-2310(03)00129-8, 2003.
Kuklinska, K., Wolska, L., and Namiesnik, J.: Air quality policy in the US and the EU–a review, Atmospheric Pollution Research, 6, 129–137, 2015.

Lelieveld, J., Evans, J., Fnais, M., Giannadaki, D., and Pozzer, A.: The contribution of outdoor air pollution sources to premature mortality on a global scale, Nature, 525, 367–371, 2015.

Lim, S. S., Vos, T., Flaxman, A. D., Danaei, G., Shibuya, K., Adair-Rohani, H., AlMazroa, M. A., Amann, M., Anderson, H. R., Andrews, K. G., et al.: A comparative risk assessment of burden of disease and injury attributable to 67 risk factors and risk factor clusters in 21 regions, 1990–2010: a systematic analysis for the Global Burden of Disease Study 2010, The lancet, 380, 2224–2260, 2013.

McKeen, S., Chung, S., Wilczak, J., Grell, G., Djalalova, I., Peckham, S., Gong, W., Bouchet, V., Moffet, R., Tang, Y., et al.: Evaluation of several PM2. 5 forecast models using data collected during the ICARTT/NEAQS 2004 field study, Journal of Geophysical Research: Atmospheres, 112, 2007.

Pozzer, A., de Meij, A., Pringle, K. J., Tost, H., Doering, U. M., van Aardenne, J., and Lelieveld, J.: Distributions and regional budgets of aerosols and their precursors simulated with the EMAC chemistry-climate model, Atmos. Chem. Phys., 12, 961–987, doi: 10.5194/acp-12-961-2012, http://www.atmos-chem-phys.net/12/961/2012/, 2012.

Webb, J., Ryan, M., Anthony, S., Brewer, A., Laws, J., Aller, M., and Misselbrook, T.: Cost-effective means of reducing ammonia emissions from UK agriculture using the NARSES model, Atmos. Environ., 40, 7222–7233, 2006.

---

## Author Comment (AC2) · 15 Aug 2017

We thank the reviewer for her/his positive comments.

1. **Please state if the ammonia reductions in this study are feasible, especially for different regions**.
   This has been mentioned in the manuscript (Page 7, line 16-20). The abatement processes for ammonia emissions are numerous and with different efficiencies. As shown by Webb et al. (2006), ammonia from livestock production (accounting for 75% of European anthropogenic emissions of ammonia) can be reduced between 20% to 80% depending on the technique adopted. It is shown that slurry stores emissions can be reduced by 80% if a solid roof, tend or lid is applied to

the storage, while an abatement efficiency between 30 to 80% can be achieve by different techniques following spreading of livestock manures to land. It it therefore reasonable to assume that the 50% reduction in agricultural ammonia emissions can be achieved with existing technology, although unclear is the real cost-efficiency of such reduction. Following Webb et al. (2006), who studied different reduction methods for the UK "[...]While there is scope for reduction in $NH_3$ emissions at moderate cost, to achieve large ($> 25\%$) reductions costs are likely to increase exponentially." A further study in this direction is under preparation, to estimate the costs of such strong reduction. The text in the manuscript will be extended and this point will be clarify (as requested also from referee #1).

2. **Section 3.2, the aerosol pH would be determined by aerosol water, which also depends on the secondary nitrate and sulfate concentrations, relative humidity etc. Further, rich or poor ammonia in different regions should have markedly different effects on aerosol pH. Please have some discussions on them.**
Indeed aerosol water plays a major role in determining the aerosol pH (Guo et al., 2015; Hennigan et al., 2015). As noticed by the referee, rich or poor ammonia in different regions have different effects on aerosol pH. This has also been mentioned in the manuscript (Page 10 line 20-22), where the large ammonia emissions from South Asia (one of the largest ammonia emitter worldwide) have strong role in reducing sulfate and nitrate. Once agricultural (and hence ammonia) emissions are reduced, the pH decreased drastically (up 1.72 pH-unit for a 100% reduction).

3. **The epidemiological studies did find the secondary inorganic aerosols could have negligible effects on human health.**
Indeed, the referee is correct, although our sentence "it is expected that some aerosol components may be more toxic than others" does not affirm the contrary. We thank the referee for the references suggested which will be implemented

in the revised manuscript, adding that secondary organic aerosols could have negligible effects on human health.

**References**

Guo, H., Xu, L., Bougiatioti, A., Cerully, K. M., Capps, S. L., Hite Jr, J., Carlton, A., Lee, S., Bergin, M., Ng, N., et al.: Fine-particle water and pH in the southeastern United States, Atmos. Chem. Phys., 15, 5211–5228, 2015.

Hennigan, C., Izumi, J., Sullivan, A., Weber, R., and Nenes, A.: A critical evaluation of proxy methods used to estimate the acidity of atmospheric particles, Atmos. Chem. Phys., 15, 2775–2790, 2015.

Webb, J., Ryan, M., Anthony, S., Brewer, A., Laws, J., Aller, M., and Misselbrook, T.: Cost-effective means of reducing ammonia emissions from UK agriculture using the NARSES model, Atmos. Environ., 40, 7222–7233, 2006.

---

## Author Response (AR1)

Dear Dr. Qiang Zhang,

here we have listed the changes applied to the manuscript, following the referees' suggestions. To facilitate the comparison with the ACPD published version, the text modifications are highlighted in the manuscript appended to this letter (see last pages). In this letter, we summarized the changes keeping the same order of the reply, although the page/line number do not correspond to the revised version.

Following the revision of the manuscript, we noticed an error in the naming and description of the simulation, as the 66% reduction simulation was actually wrong. Instead a reduction of 75% was applied in the model, and this has been corrected through the entire manuscript. This does not affect the conclusions.

**Correction made following comments of referee # 1**

1. Presenting the global perspective is interesting but I personally doubt that such work has any implications on regional policy as it entirely misses discussion of regionally specific aspect of mitigation opportunities analyzing rather unrealistic scenarios of agricultural emissions; additionally referring to 2010 levels while emissions from livestock and arable land production (fertilizer use) are likely to increase further in several regions, especially in Asia.

We extended end of Sect.2 to include the feasibility of the ammonia reduction by agricultural emissions.

2. Another aspect of this work that needs more clarity is the issue of temporal distribution of agricultural emissions used in the simulations.

We extended the discussion, moving the related text in Sect.2, where the model is evaluated, explaining the issue in USA emissions.

3. another issue is the spatial resolution of the modelled PM2.5 concentrations and its use for calculation of population exposure.

We added the discussion on model resolution in Sect.3.3

Other comments:

ABSTRACT: I am not sure if the last sentence about the impact of 100% reduction is of any significance; such reductions are not even theoretically possible.

Following the discussion we decided to keep simulation  $REF_{-100}$  as important from the scientific point of view, although not realistically achievable.

- Page 1, Line 23: One could add there a reference to the EU policy which includes now targets for  $NH_3$  emissions within the revised air quality legislation. The authors include a reference to that later in the paper. The text has been expanded.
- Page 2, Line 4: not clear what is meant by manure processing , suggest replacing with manure storage and on field application

We reformulated the sentence.

- Page 2, line 4: suggest add N or nitrogen before fertilizer The change has been implemented.
- Page 2, line 12: maybe leads should be replaced with would lead or could lead as this is a modelling study rather than impact observed anywhere. The text has been changed.
- Page 2, line 18: by agriculture should be replaced with from agriculture and resulted can be possibly modified to would or could result

Changed following referee's suggestion.

Page 2, last paragraph: As before, suggest adding a reference to the recent European air quality policy and possibly underlying analysis.

Additional text and reference to European air quality policy has been added here.

Page 3, from line 21: The emissions are for the year 2010 but the references are for data sets until 2005. Few words of explanation?

The text has been extended clarifying that the original references were for the year 2005.

- Page 4, Figure 2: A bit small, hard to read the axis The axis label were increased.
- Page 5, last paragraph: I believe it would be beneficial to put these assumptions in perspective of what has been discussed as feasible since the reductions given here, even the lowest level, are in most regions perceived as either infeasible or close to maximum reduction potential unless dietary changes are considered reducing meat demand.

We have extended the text at the end of the section.

Page 6; first paragraph: Presumably the first sentence refers to agricultural burning and so it could be moved to the end of this paragraph where combustion emissions are mentioned. In general this paragraph should be clear as to which sources are meant next to specific pollutants. The text has been modified.

Page 6; line 17: The 20-90% reduction refers to single measures and not to the overall mitigation potential and so nowhere 90% can be achieved for the whole agriculture. The potential is typically between 20-45% with some exceptions where structure is different, [..] The text has been modified accordingly to the main concerned raised

by the referee. Page 8, para from line 15: Total emissions in winter are not higher

than in summer!  $NH_3$  emissions are increasing with temperature and also organic fertilizers are applied in Spring, Summer, Autumn, just as the mineral fertilizers.

We removed the wrong text and moved the discussion on seasonality on Sect.2 (where the model evaluation is discussed).

Figure 7: here the resolution for the mortality attributable to  $PM_{2.5}$  is indicated as 10x10km. An explanation what data are used to develop that is needed. In general some discussion related to how coarse resolution concentration fields are used in health impact assessment would be useful [...]

The text has been extended, mentioning the issue related to the model coarse resolution.

**Correction made following comments of referee # 2**

- Please state if the ammonia reductions in this study are feasible, especially for different regions. The text has been extended at the end of Sect.2, following also referee's #1 comments.
- 2. Section 3.2, the aerosol pH would be determined by aerosol water, which also depends on the secondary nitrate and sulfate concentrations, relative humidity etc. Further, rich or poor ammonia in different regions should have markedly different effects on aerosol pH. Please have some discussions on them.

We believe this point to be already addressed in the manuscript, as all Sect.3.2 is dedicated to this argument. Therefore, no text was added.

3. The epidemiological studies did find the secondary inorganic aerosols could have negligible effects on human health. The text was augmented by adding the new references. Nevertheless we would like to point out that both references suggest a strong health effect by sulfate (i.e. secondary inorganic aerosols) and therefore not in agreement with the point raised by the referee. We decided to leave the text unchanged, so to be coherent with the citations.

Best regards, Andrea Pozzer (on behalf of all co-authors)

**Impact of agricultural emission reductions on fine particulate matter and public health.**

Andrea Pozzer1, Alexandra P. Tsimpidi1, Vlassis A. Karydis1, Alexander de Meij2,\*, and Jos Lelieveld1,3

1Max Planck Institute for Chemistry, Mainz, Germany

[revised manuscript text omitted]
 sheare the surface and East Asia based to global satellite for North America and East Asia (See Fig.2).

**Figure 2.** Scatter plot of observed and modeled yearly averaged concentrations of  $PM_{2.5}$  (in  $\mu g/m^3$ ). The colors denote the regions, i.e., blue North America, green Europe, purple East-South Asia and red East Asia. Black are locations outside these regions.

Further, sulfate-ammonium-nitrate has been compared with station observations from different databases, such as from EPA (United States Environmental Protection Agency), EMEP (European Monitoring and Evaluation Programme) and EANET